# Association between HLA-C alleles and COVID-19 severity in a pilot study with a Spanish Mediterranean Caucasian cohort

Lorena Vigón[1]*, Miguel Galán[1], Montserrat Torres[1], Antonio J. Martín-Galiano[2], Sara Rodríguez-Mora[1,3], Elena Mateos[1,3], Magdalena Corona[4], Rosa Malo[5], Cristina Navarro[6], María Aránzazu Murciano-Antón[7], Valentín García-Gutiérrez[4], Vicente Planelles[8], Jorge Martínez-Laso[9], María Rosa López-Huertas[1,3], Mayte Coiras[1,3]*, on behalf of the Multidisciplinary Group of Study of COVID-19 (MGS-COVID)[¶]

1 Immunopathology Unit, National Center of Microbiology, Instituto de Salud Carlos III, Madrid, Spain, 2 Intrahospital Infections Laboratory, National Centre of Microbiology, Instituto de Salud Carlos III, Madrid, Spain, 3 Biomedical Research Center Network in Infectious Diseases (CIBERINFEC), Madrid, Spain, 4 Hematology Service, Hospital Universitario Ramón y Cajal, Madrid, Spain, 5 Neumology Service, Hospital Universitario Puerta de Hierro, Majadahonda, Spain, 6 Neumology Service, Hospital de El Escorial, El Escorial, Spain, 7 Family Medicine, Centro de Salud Doctor Pedro Laín Entralgo, Alcorcón, Spain, 8 Division of Microbiology and Immunology, University of Utah School of Medicine, Salt Lake City, Utah, United States of America, 9 Immunogenetic Unit, National Center of Microbiology, Instituto de Salud Carlos III, Madrid, Spain

¶ The complete membership of the author group can be found in the Acknowledgments.
* mcoiras@isciii.es (MC); lvhernandez@isciii.es (LV)

**Data Availability Statement:** All relevant data are within the article and its Supporting Information files.

## Abstract

The clinical presentations of COVID-19 may range from an asymptomatic or mild infection to a critical or fatal disease. Several host factors such as elderly age, male gender, and previous comorbidities seem to be involved in the most severe outcomes, but also an impaired immune response that causes a hyperinflammatory state but is unable to clear the infection. In order to get further understanding about this impaired immune response, we aimed to determine the association of specific HLA alleles with different clinical presentations of COVID-19. Therefore, we analyzed HLA Class I and II, as well as KIR gene sequences, in 72 individuals with Spanish Mediterranean Caucasian ethnicity who presented mild, severe, or critical COVID-19, according to their clinical characteristics and management. This cohort was recruited in Madrid (Spain) during the first and second pandemic waves between April and October 2020. There were no significant differences in HLA-A or HLA-B alleles among groups. However, despite the small sample size, we found that HLA-C alleles from group C1 HLA-C*08:02, -C*12:03, or -C*16:01 were more frequently associated in individuals with mild COVID-19 (43.8%) than in individuals with severe (8.3%; p = 0.0030; pc = 0.033) and critical (16.1%; p = 0.0014; pc = 0.0154) disease. C1 alleles are supposed to be highly efficient to present peptides to T cells, and HLA-C*12:03 may present a high number of verified epitopes from abundant SARS-CoV-2 proteins M, N, and S, thereby being allegedly able to trigger an efficient antiviral response. On the contrary, C2 alleles are usually poorly expressed on the cell surface due to low association with β2-microglobulin (β2M) and peptides, which may impede the adequate formation of stable HLA-C/β2M/peptide

**Funding:** This work was supported by the Coordinated Research Activities at the Centro Nacional de Microbiología (CNM, Instituto de Salud Carlos III) (COV20_00679) to promote an integrated response against SARS-CoV-2 in Spain (Spanish Ministry of Science and Innovation) that is coordinated by Dr Inmaculada Casas (WHO National Influenza Center of the CNM); donation provided by Chiesi España, S.A.U. (Barcelona, Spain); the Spanish Ministry of Science and Innovation (PID2019-110275RB-I00); and grant MPY509/19 provided by Instituto de Salud Carlos III. The work of MRLH and SRM is financed by NIH grant R01AI143567. The work of LV is supported by a pre-doctoral contract from Instituto de Salud Carlos III (FIS PI16CIII/00034-ISCIII-FEDER). The work of MT is supported by Instituto de Salud Carlos III (COV20_00679). AJMG is the recipient of a post-doctoral contract "Miguel Servet" supported by the Instituto de Salud Carlos III.

**Competing interests:** The authors have declared that no competing interests exist.

heterotrimers. Consequently, this pilot study described significant differences in the presence of specific HLA-C1 alleles in individuals with different clinical presentations of COVID-19, thereby suggesting that HLA haplotyping could be valuable to get further understanding in the underlying mechanisms of the impaired immune response during critical COVID-19.

## Introduction

In 2019, the severe acute respiratory syndrome (SARS) coronavirus type 2 (SARS-CoV-2) emerged, becoming the etiological agent of the major pandemic in this century. More than 296 million people have passed the coronavirus disease 2019 (COVID-19) worldwide so far, with approximately 5.4 million deaths [1]. These numbers are rapidly increasing due to new emergent variants such as Omicron [2]. Not all the individuals respond similarly to SARS-CoV-2 infection and, therefore COVID-19 may have different presentations, ranging from an asymptomatic infection to a critical or fatal disease [3]. Although it is still uncertain why some individuals develop the most severe forms of the disease, an anomalous innate immune response that ends in a cytokine storm seems to be a major cause for a poor prognosis [4,5]. Accordingly, high levels of interleukin (IL)-7, IL-6, ferritin, or C reactive protein (CRP) have been detected in the plasma of individuals with severe COVID-19 [6,7]. On the contrary, these patients usually show low levels of cytokines related to T-cell antiviral activity, such as IL-2, IL-12 or interferon (IFN)-γ [8], as well as reduced antiviral responses mediated by IFN type-I and -III [9,10]. Consequently, this defective immune response would not only be unable to clear the infection by SARS-CoV-2, but patients may even experience the reactivation of latent herpesvirus infections [11]. For a long time, the only treatment available for the hyper inflammation state derived from SARS-CoV-2 infection has been corticosteroids such as dexamethasone [12], but recently, the Food and Drug Administration (FDA) has approved the oral administration of nirmatrelvir and ritonavir for the emergency treatment of mild-to-moderate COVID-19 in children and adults who present high risk for progression to severe COVID-19, as well as oral molnupiravir for adults [13,14].

Within the host factors that may influence the susceptibility to develop the most severe forms of COVID-19, the human leukocyte antigen (HLA) complex has been appointed as a potential candidate [15]. The HLA complex is formed by highly polymorphic genes located on chromosome 6 that encode glycoproteins that regulate the immune response by presenting exogenous and endogenous peptides to T lymphocytes and Natural Killer (NK) cells [16]. HLA genes are inherited as a group known as haplotype [17]. Classical HLA class I glycoproteins A, B, and C are expressed on the surface of all nucleated cells in association with the invariant β2 microglobulin (β2M), a protein encoded in chromosome 15 that stabilizes the tertiary structure of these molecules [18,19]. On the other hand, HLA class II glycoproteins DR, DQ, DM, DO, and DP are located only on the surface of professional antigen presenting cells (APCs) such as macrophages, B cells and dendritic cells [20]. HLA class I glycoproteins mostly present intracellular small peptides to CD8+ T cells, such as those derived from a viral infection [18,21], whereas HLA class II glycoproteins mostly present exogenous peptides to CD4 + T cells, which would induce a humoral response through the activation of specific B cells [22,23]. HLA-C molecules are also recognized by NK cells, thereby acting mostly as ligands for the Killer Immunoglobulin-like Receptors (KIRs) that are expressed on the cell surface [24] and participate in the activation or inhibition of these cells [25].

An efficient presentation of peptides by HLA molecules to the cells of the immune system is essential to initiate a potent and effective response against pathogens such as SARS-CoV-2 in

order to swiftly clear the infection. However, due to the high polymorphism of HLA, not all molecules are equally effective to generate an efficient response. More than 24,000 HLA alleles have been reported so far, mostly within HLA class I [26]. In the Spanish population, HLA-B shows the greatest allele number, whereas HLA-A and -C show the major polymorphism [27]. HLA-B and -C alleles are hierarchically classified according to the amino acid residues located at positions 77 and 80–83 of the peptide-binding pocket. Therefore, HLA-B is traditionally subdivided into -Bw6 (Ser[77] and Asn[80]) and -Bw4 epitopes [28] that are characterized by at least seven different patterns of amino acid exchanges in these positions, whereas HLA-C alleles are subdivided into groups C1 and C2, which are mutually exclusive. Alleles from group C1 contain Ser[77] and Asn[80] and are ligands for the inhibitory KIR2DL2 and -2DL3, as well as for the activating KIR2DS2, whereas alleles from group C2 contain Asn[77] and Lys[80] [29] and are ligands for the inhibitory KIR2DL1 and activating KIR2DS1 [30]. As a major characteristic, HLA-C molecules show a distinctly lower expression on the cell surface in comparison with HLA-A and -B molecules, most likely due to low mRNA expression, poor assembly with β2M, or restricted peptide binding that leads to the accumulation of misfolded intermediates in the endoplasmic reticulum [31].

The potential role of HLA molecules in the different susceptibility to infectious diseases has been described for several alleles [32,33]. In fact, the expression of specific HLA-B and -C alleles have been associated with higher resistance to infection by the human immunodeficiency virus type 1 (HIV-1) [34–36] and with the spontaneous clearance of hepatitis C virus (HCV). Moreover, the presence or absence of specific HLA class I and II molecules appear to be essential for the immune response against respiratory infections such as tuberculosis [37,38], influenza [39], or the first SARS-CoV. In the latter case, the expression of specific HLA alleles such as HLA-B*07:03 and -B*46:01 may confer a higher predisposition to develop severe SARS [40,41], whereas the presence of HLA-C*15:02 may have a protective effect [42]. Regarding the specific association of HLA alleles with COVID-19, healthy individuals carrying HLA-B*44 and/or -C*01 (group C1) may be more susceptible to be infected with SARS-CoV-2 and to spread the infection within the population [43]. Other HLA-C alleles from group HLA-C1 have been described to show a higher degree of recognition of SARS-CoV-2 peptides [44], and HLA-C*01 has also been correlated with a higher mortality during COVID-19 [45] in the Spanish population.

During the first COVID-19 pandemic peak (April to May 2020), Spain was one of the European countries more severely affected, with approximately 100 excess deaths per 100,000 people [46] and a case fatality rate 2.3% higher than the average rate in the neighboring countries [47]. The causes for this nearly 38% of relative increased death have not been completely elucidated yet. Therefore, in this pilot study, we analyzed the potential association between the different clinical presentations of COVID-19 and the presence of specific HLA and KIR alleles in a cohort of Spanish Mediterranean Caucasian individuals who were recruited during the first wave of SARS-CoV-2. This information could be useful to identify predictive biomarkers for COVID-19 severity and to further understand the underlying pathogenesis of this emergent disease.

## Materials and methods

### Patients

Blood samples were obtained from 72 individuals with Spanish ethnicity (Mediterranean Caucasian origin) who developed different presentations of COVID-19. All patients were recruited in Madrid (Spain) after the first pandemic peak and at the beginning of the second wave, between April and October 2020. The inclusion criteria were to have a positive SARS-CoV-2

qRT-PCR assay in nasopharyngeal smear and to be older than 18 years old at the moment of sampling. Twenty-four participants were asymptomatic or showed mild symptoms of COVID-19 and hospitalization was not required, whereas 48 patients were hospitalized due to the severity of the symptoms. Hospitalized patients were classified as severe (n = 17) or critical (n = 31), based on COVID-19 evolution during hospitalization. Patients with critical COVID-19 were those who needed intensive care, invasive mechanical ventilation, and/or died due to complications of the disease. Main demographic and clinical characteristics of all individuals recruited for this study are summarized in Table 1.

## Ethical statement

The individuals with the most severe forms of COVID-19 were recruited from Hospital Universitario Ramón y Cajal, Hospital Universitario Puerta de Hierro, and Hospital de El Escorial (Comunidad de Madrid, Spain). Participants with mild COVID-19 were recruited at the Primary Healthcare Center Doctor Pedro Lain Entralgo (Alcorcón, Madrid, Spain). All patients gave informed written consent to participate in the study or witnessed oral consent with written consent by a representative to avoid handling contaminated documents. The confidentiality and anonymity of all participants were secured by Current Spanish and European Data Protection Acts. Protocol for this study (CEI PI 32_2020-v2) was prepared in accordance with the Helsinki Declaration and previously reviewed and approved by the Ethics Committees of Instituto de Salud Carlos III (IRB IORG0006384) and all participating hospitals.

## DNA isolation and HLA genotyping

Whole genomic DNA was isolated from blood using QIAamp DNA Blood Mini Kit (Qiagen Iberia, Madrid, Spain). Intermediate HLA genotyping resolution (four digits, the corresponding "P" for every $2^{nd}$ field allele typing was discarded to clarify the text) was performed with 2μg of genomic DNA using LinkSēq™ HLA ABCDRDQB1 384 kit (One Lambda, Thermo Fisher Scientific España, Madrid, Spain), which allows testing human class I classical HLA-A, -B, -C alleles and class II–DR, -DQ and–DP alleles, including Bw4 and Bw6 genotypes. qPCR was performed using QuantStudio 5 Real-Time PCR System (Thermo Fisher Scientific),

**Table 1. Demographic and clinical data of participants with different presentations of COVID-19 who were recruited for the study.**

|  | Mild | Severe | Critical |
|---|---|---|---|
| **Male/Female (n)** | 10/14 | 11/6 | 20/11 |
| **Median age (years)** | 47.5 (IQR 32.8–58.8) | 72.0 (IQR 68.0–86.5) | 65.0 (IQR 58.0–72.0) |
| **Days from clinical onset to sample** | 88.5 (IQR 79.8–95.0) | 23.0 (IQR 9.3–39.3) | 35.0 (IQR 20.0–53.0) |
| **Median length of hospitalization (days)** | N/A | 17.0 (IQR 8.5–28.5) | 48.0 (IQR 32.0–81.0) |
| **Median length at the ICU (days)** | N/A | N/A | 43.5 (IQR 21.3–59.0) |
| **Pneumonia (Yes/No/UD)** | 2/21/1 | 14/3/0 | 26/3/2 |
| **Invasive mechanical ventilation (Yes/No/UD)** | 0/24/0 | 1/16/0 | 25/5/1 |
| **DIC (Yes/No/UD)** | 0/24/0 | 5/12/0 | 1/29/1 |
| **Dyslipidemia (Yes/No/UD)** | 6/18/0 | 5/12/0 | 14/15/2 |
| **Diabetes (Yes/No/UD)** | 1/23/0 | 4/13/0 | 1/27/3 |
| **Hypertension (Yes/No/UD)** | 4/20/0 | 10/7/0 | 14/15/2 |
| **Exitus (Yes/No)** | 0/24 | 0/17 | 14/17 |

DIC, Disseminated intravascular coagulation; ICU, Intensive Care Unit; IQR: Interquartile range; N/A. Not applicable; UD, Undetermined.

according to manufacturer's instructions and using the following conditions: 95°C, 2 min; 36 cycles: 95°, 15 seconds; 64°C, 1 min. A dissociation profile was added using the following conditions: 95°C, 15 seconds; 65°C, 30 seconds. The dissociation data corresponding to the melt curve were collected per well between 65°C and 95°C using QuantStudio™ Design & Analysis Software (Thermo Fisher Scientific). These data were exported to SureTyper software (One Lambda) for the interpretation and identification of HLA alleles.

## HLA haplotyping

Statistical analysis was performed with Arlequin v3.5 software [48]. In brief, this program calculates HLA allele frequencies, Hardy-Weinberg equilibrium (HWE), which assumes random mating within the population, and the linkage disequilibrium between n alleles at n different loci and their level of significance (p) for 2x2 comparisons to establish the 2, 3 and 4 loci associations. In addition, the most frequent complete extended haplotypes were deduced from: 1) 2-loci HLA haplotype frequencies, 2) from the previously described haplotypes in other populations; and 3) from the haplotypes if they appeared in two or more individuals and the alternative haplotype was well defined [49].

## KIR genotyping

KIR genotyping was performed with 360 ng of genomic DNA using LinkSeq KIR Typing kit (One Lambda, Thermo Fisher Scientific). This kit allows testing all 15 human KIR genes and 2 pseudogenes, including both full length and deleted forms of 3DP1 and 2DS4 and allele-specific variants in 3DL1, 2DL1. qPCR was performed using StepOne Real-Time PCR System (Thermo Fisher Scientific) according to manufacturer's instructions and using the following conditions: 95°C, 2 min; 36 cycles: 95°, 15 seconds; 64°C, 1 min. A dissociation profile was added (95°C, 15 seconds; 65°C, 30 seconds) and dissociation data corresponding to the melt curve of the amplification were collected per well between 65°C and 95°C using StepOne Plus software v2.3. These data were exported to SureTyper software (One Lambda) for interpretation and identification of KIR genotypes.

## HLA class I epitope prediction and analysis

HLA class I epitopes between 8–12 residues for the whole reference SARS-CoV-2 proteome (Wuhan-1; RefSeq: NC_045512.2) were predicted for all HLA class I alleles observed in the patient cohort using the NetMHCpan-4.1 server that may predict the binding of the viral peptides to any known MHC molecule using artificial neural networks [50]. Binding epitopes were considered those with rank $\leq$ 0.5 and score $\geq$ 0.5. Redundant epitopes for the same allele and sharing the same peptide core but distinct total lengths and lower scores, were ignored. Experimentally validated epitopes were downloaded from the Immune Epitope Database and Analysis Resource (Last accession: 19/03/2021) [51], using the following search terms: Epitopes: "Any epitopes"; Assay: "T Cell", "MHC Ligand" and outcome: "Positive"; MHC Restriction: "MHC Class I"; Host: "Human"; Disease "COVID-19 [52]". Predicted epitopes were considered as validated when there was a validated epitope that shared perfect matching protein coordinates. Distribution of HLA alleles in mild, severe and critical COVID-19 was represented with GraphPad Prism software (GraphPad Software Inc.).

## HLA-C alleles 3D modelling

The three-dimensional (3D) structure of HLA-C*07:02 was obtained from the Protein Data Bank (PDB) (PDB code: 6PA1) [53] to be used as a template, and it was visualized using the

PyMOL Molecular Graphics System, 2.0 (Schrödinger, LLC). Mutagenesis in silico was performed in this template to obtain model 3D structures for consensus sequences, including key pocket residues of HLA-C*08:02 (group C1) and HLA-C*07:02 (group C2) alleles in order to determine the modifications produced in the binding pocket due to amino acid exchanges. Amino acids at positions 9, 11, 24, 77, 80, 90, 94, and 95 were selected as key residues in the structure of HLA-C pockets and the amino acid sequences of HLA-C alleles, as well as the corresponding multiple alignments, were obtained from the Immuno Polymorphism database (IPD)- Immunogenetics (IMGT)/HLA Database (EMBL-EBI, Wellcome Genome Campus, Hinxton, UK) [26].

## Statistical analysis

Statistical significance between groups was determined by comparing the data obtained in the groups of individuals with severe and critical presentations of COVID-19 with the data obtained from the individuals who had mild or asymptomatic COVID-19. Categorical data were expressed as percentages, and the median was used to describe tendency. Statistical significance for two-row by two-column (2x2) contingency table was determined by Fisher's exact test or chi-square test using Graph Pad Prism 8.0 (Graph Pad Software Inc., San Diego, CA). Alleles with counts less than five in every group were combined into a common group before computing the test. Significance levels were corrected by Bonferroni correction. For each HLA haplotype the corrected P value (Pc) was calculated by multiplying the P value obtained with the two-tailed chi-square or Fisher's exact test by the number of tested allelic combinations. pc<0.05 were considered statistically significant in all comparisons.

## Results

### Patients' characteristics

Seventy-two patients of Spanish Mediterranean Caucasian origin who were diagnosed with COVID-19 by positive qRT-PCR were recruited for this study between April and October 2020. The most relevant demographic and clinical characteristics of all participants are summarized in Table 1 and all detailed characteristics for non-hospitalized and hospitalized individuals are shown in S1 and S2 Tables, respectively. The individuals were classified into three groups depending on the medical assistance they received and according to the progression of the disease. Twenty-four patients received primary health assistance from Primary Healthcare Centers, and they were homebound until the resolution of the symptoms or a negative qRT-PCR test for SARS-CoV-2 (henceforth, Mild COVID-19). 58.3% of these individuals were females with a median age at diagnosis of 47.5 years (IQR 32.8–58.8). Median days from clinical onset to sample acquisition was 88.5 days (IQR 79.8–95.0). Forty-eight patients were hospitalized due to COVID-19 severity. Of these individuals, 31 were categorized as critical and admitted to the Intensive Care Unit (ICU) (henceforth, Critical COVID-19), whereas the rest of hospitalized individuals (n = 17) presented severe forms of COVID-19 but without requiring admission to the ICU (henceforth, Severe COVID-19). Most severe and critical patients were males (64.7% and 64.5%, respectively), with median age at diagnosis of 72.0 (IQR 68.0–86.5) and 65.0 (IQR 58.0–72.0) years, respectively. Median number of days from clinical onset to sample acquisition was 23.0 (IQR 9.3–39.3) and 35.0 (IQR 20.0–53.0) days in severe and critical, respectively. Median length of hospitalization was 17.0 (IQR 8.5–28.5) and 48.0 (IQR 32.0–81.0) days in severe and critical patients, respectively. Median length of hospitalization at the ICU was 43.5 days (IQR 21.3–59.0) in critical patients. Fourteen individuals with critical COVID-19 (45.2%) expired due to complications of the disease. The most common symptoms showed by individuals with mild COVID-19 were asthenia (75%), fever

(58.3%), and cough (58.3%). Most individuals with mild COVID-19 did not develop pneumonia (87.5%), whereas pneumonia (82.4%) and cough (52.9%) were the most frequent symptoms in severe patients and pneumonia (83.9%), fever (74.2%), and cough (67.7%) were the most frequent symptoms in critical individuals. Among the potential risk factors for developing severe forms of COVID-19, dyslipidemia was present in 25.0%, 29.4%, and 45.2% of individuals with mild, severe, and critical COVID-19, respectively, whereas arterial hypertension was present in 16.7%, 64.7%, and 45.2%, respectively.

## HLA allele frequencies

HLA class I and II allele frequencies were analyzed in individuals with mild, severe and critical COVID-19 and the statistical significance of the comparison between the groups with severe and critical COVID-19 and the group with mild COVID-19 was calculated by Fisher's exact test or chi-square test 2x2 and corrected by Bonferroni correction (Table 2). The most common HLA-C alleles in our cohort were HLA-C*04:01 and -C*05:01, but no significant differences were found for the frequency of these alleles in the comparison between groups. However, the presence of HLA-C*08:02 was significantly higher in individuals with mild COVID-19 (18.8%) in comparison with individuals with critical COVID-19 (1.6%; p = 0.0024; pc = 0.0240). The most common HLA-B alleles in our cohort were HLA-B*44:03 and -B*18:01, but no significant differences were found for the frequency of these alleles between groups. The frequency of HLA-B rare alleles was higher in individuals with severe COVID-19 (82.4%) regarding those with mild COVID-19 (54.2%; p = 0.0080; pc = 0.0400), but no significance in the comparison between mild and critical COVID-19 was achieved. The most frequent HLA-A allele in our cohort was HLA-A*02:01, whereas the most frequent HLA class II alleles were DRB1*07:01 and *03:01, but no significant differences were observed in the comparison between groups for these alleles.

The frequency of associations between specific HLA-A, -B and -C alleles that may interact with KIRs was also analyzed and the statistical significance of the comparison between the groups with severe and critical COVID-19 and the group with mild COVID-19 was calculated by Fisher's exact test or chi-square 2x2 (Table 3). The frequency of associations between HLA-C*08 or -C*12 alleles was significantly higher in individuals with mild COVID-19 (33.3%) than in individuals with critical COVID-19 (8.1%) (p = 0.0008; pc = 0.0088). Moreover, specific HLA-C*08:02 or -C*12:03 alleles were more frequent in 31.3% of individuals with mild COVID-19, in comparison with 3.2% (p = 0.0001; pc = 0.0010) of individuals with critical disease. These differences were still significant in the comparison between groups when the frequency of the association between alleles HLA-C*08:02, HLA-C*12:03 or HLA-C*16:01 was analyzed, being more frequent in individuals with mild COVID-19 (43.8%) than in individuals with severe (8.3%; p = 0.0030; pc = 0.0330) or critical COVID-19 (16.1%; p = 0.0014; pc = 0.0154).

## HLA haplotype frequencies

HLA complex is highly stable and allows little recombination between genes. Therefore, the most frequent HLA haplotypes were deduced from the two HLA loci haplotype frequencies [49]. The extended HLA haplotypes that were more frequent in our cohort are summarized in Table 4. No significant differences were observed in the comparison between groups. Although we also observed a different haplotype pattern in the frequency of HLA-B genotypes -Bw4 and -Bw6 amongst the different groups of patients, these results did not reach statistical significance between the groups (S1 Fig).

**Table 2. HLA class I and II allele frequencies in individuals with mild, severe and critical COVID-19 who were recruited for this pilot study.** The statistical significance of the comparisons between the allele frequencies of groups of individuals with severe and critical COVID-19 and the group of individuals with mild COVID-19 was calculated using Fisher's exact test or chi-square test 2x2. Significance levels were corrected by Bonferroni correction. Significant pc <0.05 are highlighted in bold font.

| | Mild COVID-19 | Severe COVID-19 | Critical COVID-19 | Mild *vs* Severe p-value | Mild *vs* Critical p value | Mild *vs* Severe pc-value | Mild *vs* Critical pc value |
|---|---|---|---|---|---|---|---|
| | **n = 48** | **n = 34** | **n = 62** | | | | |
| *Class I* | **n (%)** | **n (%)** | **n (%)** | 0.0857 | 0.4099 | ns | ns |
| *A*02:01* | 15 (31.3) | 5 (14.7) | 15 (24.2) | 0.6883 | 0.5087 | ns | ns |
| *A*03:01* | 3 (6.3) | 3 (8.8) | 7 (11.3) | 1 | 0.1346 | ns | ns |
| *A*11:01* | 1 (2.1) | 1 (2.9) | 7 (11.3) | 0.1463 | 1 | ns | ns |
| *A*23:01* | 4 (8.3) | 0 (0) | 5 (8.1) | 0.7669 | 1 | ns | ns |
| *A*24:02* | 7 (14.6) | 6 (17.6) | 8 (12.9) | 1 | 1 | ns | ns |
| *A*29:02* | 4 (8.3) | 2 (5.9) | 5 (8.1) | 0.0553 | 0.5571 | ns | ns |
| *Rare alleles* | 14 (29.2) | 17 (50.0) | 15 (24.2) | 1 | 0.4619 | ns | ns |
| *B*07:02* | 2 (4.2) | 1 (2.9) | 6 (9.7) | 0.0378 | 0.0203 | 0.1890 | 0.1015 |
| *B*14:01* | 7 (14.6) | 0 (0) | 1 (1.6) | 1 | 0.8980 | ns | ns |
| *B*18:01* | 5 (10.4) | 4 (11.8) | 6 (9.7) | 0.0734 | 0.2754 | ns | ns |
| *B*44:03* | 8 (16.7) | 1 (2.9) | 6 (9.7) | 0.0080 | 0.1023 | **0.0400** | ns |
| *Rare alleles* | 26 (54.2) | 28 (82.4) | 43 (69.4) | 0.6382 | 1 | ns | ns |
| *C*02:02* | 3 (6.3) | 1 (2.9) | 5 (8.1) | 0.1866 | 0.5461 | ns | ns |
| *C*04:01* | 4 (8.3) | 7 (20.6) | 8 (12.9) | 0.6937 | 0.5223 | ns | ns |
| *C*05:01* | 5 (10.4) | 2 (5.9) | 9 (14.5) | 0.0770 | 1 | ns | ns |
| *C*06:02* | 1 (2.1) | 5 (14.7) | 2 (3.2) | 0.0296 | 0.4655 | 0.2960 | ns |
| *C*07:01* | 2 (4.2) | 7 (20.6) | 5 (8.1) | 0.4411 | 0.7288 | ns | ns |
| *C*07:02* | 3 (6.3) | 4 (11.8) | 6 (9.7) | 0.0405 | 0.0024 | 0.4050 | **0.0240** |
| *C*08:02* | 9 (18.8) | 1 (2.9) | 1 (1.6) | 0.4593 | 0.0417 | ns | 0.4170 |
| *C*12:03* | 6 (12.5) | 2 (5.9) | 1 (1.6) | 0.2301 | 1 | ns | ns |
| *C*16:01* | 6 (12.5) | 1 (2.9) | 8 (12.9) | 0.5427 | 0.3672 | ns | ns |
| *Rare alleles* | 9 (18.8) | 4 (11.8) | 17 (27.4) | 1 | 0.4998 | ns | ns |
| *Class II* | **n = 48** | **n = 32** | **n = 62** | 0.2292 | 0.3664 | ns | ns |
| *DRB1*03:01* | **n (%)** | **n (%)** | **n (%)** | 0.5607 | 0.2290 | ns | ns |
| *DRB1*07:01* | 5 (10.4) | 4 (12.5) | 4 (6.5) | 0.4288 | 0.5087 | ns | ns |
| *DRB1*13:01* | 13 (27.1) | 5 (15.6) | 12 (19.4) | 0.9271 | 0.9440 | ns | ns |
| *DRB1*15:01* | 1 (2.1) | 2 (6.2) | 5 (8.1) | 0.0286 | 0.0107 | 0.1430 | 0.0642 |
| *Rare alleles* | 3 (6.3) | 4 (12.5) | 7 (11.3) | 0.2890 | 0.5223 | ns | ns |
| *DQB1*02:02* | 26 (54.2) | 17 (53.1) | 34 (54.8) | 0.7070 | 0.1744 | ns | ns |
| *DQB1*03:01* | 15 (31.8) | 3 (13.9) | 11 (14.5) | 0.6788 | 0.7288 | ns | ns |
| *DQB1*03:02* | 5 (11.4) | 6 (19.4) | 9 (14.5) | 1 | 0.4619 | ns | ns |
| *DQB1*06:02* | 4 (9.1) | 4 (13.9) | 11 (18.2) | 0.7107 | 0.3282 | ns | ns |
| *DQB1*06:03* | 3 (6.8) | 3 (8.3) | 6 (10.9) | | | | |
| *Rare alleles* | 2 (0) | 2 (0) | 6 (9.1) | | | | |
| | 19 (39.6) | 14 (43.8) | 19 (30.6) | | | | |

n: Number of alleles.

HLA-A rare alleles: *01:01; *02:02; *25:01; *26:01; *29:01; *30:01; *30:02; *32:01; *33:01; *66:01; *68:01; *68:02; *69:01.

HLA-B rare alleles: *07:01; *07:05; *08:01; *14:02; *15:01; *15:03; *15;16; *27:05; *35:01; *35:02; *35:03; *35:05; *38:01; *39:01; *39:03; *39:05; *40:01; *40:02; *40:06; *41:02; *44:02; *45:01; *47:01; *48:01; *49:01; *50:01; *50:02; *52:01; *53:01; *55:01; *57:01; *58:01; *67:01.

HLA-C rare alleles: *01:02; *03:02; *03:03; *03:04; *08:01; *12:02; *14:02; *15:02; *15:05; *16:02; *17:01.

HLA-DRB1 rare alleles: *01:01; *01:02; *01:03: *04:01; *04:02; *04:04; *04:05; *04:06; *04:11; *08:02; *08:04; *10:01; *11:01; *11:02; *11:04; *12:01; *13:02; *13:03; *14:01; *14:02; *15:02; *16:01.

HLA-DQB1 rare alleles: *02:10; *03:03; *04:02; *05:01; *05:02; *05:03; *06:01; *06:04; *06:09.

## Association between KIR genes and alleles and clinical presentations of COVID-19

The association between specific KIR genes and alleles and the different presentations of COVID-19 was analyzed. Due to the low quantity of DNA available in some samples, KIR genotyping could only be performed in 55 patients of our cohort (mild, n = 18; severe, n = 17; critical, n = 20). No significant differences were observed between the frequencies of thirty alleles

**Table 3. Frequency of the associations between specific HLA class I alleles that may interact with KIR in individuals with mild, severe and critical COVID-19.** The statistical significance of the comparisons between the frequencies of these allele combinations in the groups of individuals with severe and critical COVID-19 and the group of individuals with mild COVID-19 was calculated using Fisher's exact test or chi-square 2x2. Significance levels were corrected by Bonferroni correction. Significant pc <0.05 are highlighted in bold font.

| HLA alleles | Mild COVID-19 n = 48 | Severe COVID-19 n = 34 | Critical COVID-19 n = 62 | Mild vs Severe p-value | Mild vs Critical p-value | Mild vs Severe pc-value | Mild vs Critical pc-value |
|---|---|---|---|---|---|---|---|
| | n (%) | n (%) | n (%) | | | | |
| HLA-A*03 or HLA-A*11 | 4 (8.3) | 4 (11.8) | 14 (22.6) | 0.7125 | 0.0675 | ns | ns |
| HLA-B*14 or HLA-B*18 | 14 (29.2) | 5 (14.7) | 8 (12.9) | 0.1263 | 0.0344 | ns | ns |
| HLA-B*35 or HLA-B*49 | 3 (6.3) | 8 (23.5) | 8 (12.9) | 0.0237 | 0.3424 | 0.2607 | ns |
| HLA-C*08 or HLA-C*12 | 16 (33.3) | 3 (8.8) | 5 (8.1) | 0.0154 | 0.0008 | 0.1694 | **0.0088** |
| HLA-C*05 or HLA-C*16 | 11 (22.9) | 3 (8.8) | 18 (29.0) | 0.1375 | 0.4703 | ns | ns |
| HLA-C*06 or HLA-C*07 | 6 (12.5) | 16 (47.1) | 11 (17.7) | 0.0008 | 0.4507 | **0.0088** | ns |
| HLA-B*14:01 or HLA-B*18:01 | 12 (25.0) | 4 (11.8) | 7 (11.3) | 0.1658 | 0.0592 | ns | ns |
| HLA-C*08:02 or HLA-C*12:03 | 15 (31.3) | 3 (8.8) | 2 (3.2) | 0.0168 | 0.0001 | 0.1848 | **0.0010** |
| HLA-C*05:01 or HLA-C*16:01 | 11 (22.9) | 3 (8.8) | 17 (27.4) | 0.1375 | 0.5908 | ns | ns |
| HLA-C*06:02 or HLA-C*07:01 | 3 (6.3) | 12 (35.3) | 7 (11.3) | 0.0012 | 0.5087 | **0.0132** | ns |
| HLA-C*08:02 or HLA-C*12:03 or HLA-C*16:01 | 21 (43.8) | 4 (8.3) | 10 (16.1) | 0.0030 | 0.0014 | **0.0330** | **0.0154** |

n: Number of alleles.

of fifteen KIR loci and two pseudogenes in the groups of individuals with severe and critical COVID-19 in comparison with the group of individuals with mild COVID-19 (Table 5).

## Association between HLA-C alleles and COVID-19 presentations

Some HLA-C alleles can be divided into two large groups according to the amino acids located at positions 77 and 80 when they are considered as ligands for KIRs. Alleles from group C1 contain $Ser^{77}$ and $Asn^{80}$ (HLA-C*01; *03; *08; *12; *14, and *16), whereas alleles from group C2 contain $Asn^{77}$ and $Lys^{80}$ (HLA-C*02; *04; *05; *06; *07; *15; *17, and *18) [29]. When the frequency of hetero- and homozygosis of HLA-C1 and/or -C2 alleles was analyzed in the

**Table 4. Most common HLA haplotypes observed in individuals with mild, severe and critical COVID-19.** The statistical significance of the comparisons between the frequencies of these haplotypes in the groups of individuals with severe and critical COVID-19 and the group of individuals with mild COVID-19 was calculated using Fisher's exact test 2x2. Significance levels were corrected by Bonferroni correction. Significant pc <0.05 are highlighted in bold font.

| HLA Haplotypes | Mild COVID-19 n = 48 | Severe COVID-19 n = 34 | Critical COVID-19 n = 60 | Mild vs Severe p-value | Mild vs Critical p-value | Mild vs Severe pc-value | Mild vs Critical pc-value |
|---|---|---|---|---|---|---|---|
| | n (%) | n (%) | n (%) | | | | |
| A*x B*07:02 C*07:02 DRB1*15:01 DQB1*06:02 | 1 (2.1) | 2 (5.9) | 4 (6.7) | 0.1545 | 0.3842 | ns | ns |
| A*x B14:01 C*08:02 DRB1*07:01 DQB1*02:02 | 6 (12.6) | 0 (0) | 1 (1.7) | 0.0389 | 0.0417 | 0.2723 | 0.2919 |
| A*x B44:03 C*16:01 DRB1*07:01 DQB1*02:02 | 5 (10.5) | 1 (2.9) | 1 (1.7) | 0.3927 | 0.0840 | ns | ns |
| A*x B*44:03 C*x DRB1*07:01 DQB1*02:02 | 7 (14.7) | 1 (2.9) | 1 (1.7) | 0.1311 | 0.0203 | ns | 0.1421 |
| A*x B18:01 C*05:01 DRB1*03:01 DQB1*02:01 | 3 (6.3) | 1 (2.9) | 5 (8.3) | 0.6382 | 1 | ns | ns |
| B*14:01 C*08:02 | 7 (14.7) | 0 (0) | 1 (1.7) | 0.1311 | 0.0203 | ns | 0.1421 |
| B*44:03 C*16:01 | 6 (12.6) | 2 (5.9) | 3 (5.0) | 0.4593 | 0.1747 | ns | ns |

n: Number of alleles.

**Table 5. KIR genes and alleles identified in individuals with mild, severe and critical COVID-19.** The statistical significance of the comparisons between the frequencies of KIR genes and alleles in the groups of individuals with severe and critical COVID-19 and the group of individuals with mild COVID-19 was calculated using Fisher's exact test 2x2. Significance levels were corrected by Bonferroni correction.

| KIR genes | Mild COVID-19 n = 18 n (%) | Severe COVID-19 n = 17 n (%) | Critical COVID-19 n = 20 n (%) | Mild vs Severe p-value | Mild vs Critical p-value | Mild vs Severe pc-value | Mild vs Critical pc-value |
|---|---|---|---|---|---|---|---|
| 2DL1 | 17 (94.4) | 16 (94.1) | 20 (100) | 1 | 0.4737 | ns | ns |
| 2DL2 | 10 (55.5) | 10 (58.8) | 13 (65.0) | 1 | 0.7409 | ns | ns |
| 2DL3 | 15 (83.3) | 14 (82.4) | 19 (95.0) | 1 | 0.3282 | ns | ns |
| 2DL4 | 18 (100) | 17 (100) | 20 (100) | 1 | 1 | ns | ns |
| 2DL5 | 13 (72.2) | 10 (58.8) | 12 (60.0) | 0.4887 | 0.5064 | ns | ns |
| 2DP1 | 16 (88.9) | 16 (94.1) | 20 (100) | 1 | 0.2176 | ns | ns |
| 2DS1 | 12 (66.7) | 9 (52.9) | 7 (35.0) | 0.4998 | 0.1031 | ns | ns |
| 2DS2 | 10 (55.6) | 10 (58.8) | 11 (55.5) | 1 | 1 | ns | ns |
| 2DS3 | 5 (27.8) | 3 (17.6) | 8 (40.0) | 0.6906 | 0.5064 | ns | ns |
| 2DS4 | 18 (100) | 14 (82.4) | 19 (95.0) | 0.1039 | 1 | ns | ns |
| 2DS5 | 8 (44.4) | 8 (47.1) | 8 (40.0) | 1 | 1 | ns | ns |
| 3DL1 | 18 (100) | 15 (88.2) | 20 (100) | 0.2286 | 1 | ns | ns |
| 3DL2 | 18 (100) | 17 (100) | 20 (100) | 1 | 1 | ns | ns |
| 3DL3 | 18 (100) | 17 (100) | 20 (100) | 1 | 1 | ns | ns |
| 3DP1 | 18 (100) | 17 (100) | 20 (100) | 1 | 1 | ns | ns |
| 3DS1 | 9 (50.0) | 7 (41.2) | 7 (35.0) | 0.7380 | 0.5118 | ns | ns |

| KIR alleles | Mild COVID-19 n = 18 n (%) | Severe COVID-19 n = 17 n (%) | Critical COVID-19 n = 20 n (%) | Mild vs Severe p-value | Mild vs Critical p-value | Mild vs Critical pc-value | Mild vs Critical pc-value |
|---|---|---|---|---|---|---|---|
| 2DL1-001.1 | 17 (94.4) | 16 (94.1) | 20 (100) | 1 | 0.6062 | ns | ns |
| 2DL1-028.1 | 0 (0) | 0 (0) | 17 (85.0) | 1 | 1 | ns | ns |
| 2DL1-021.1 | 15 (83.3) | 14 (82.4) | 8 (40.0) | 1 | 1 | ns | ns |
| 2DL1-022.1 | 5 (27.8) | 4 (23.5) | 13 (65.0) | 1 | 0.5064 | ns | ns |
| 2DL2-002.1 | 10 (55.6) | 10 (58.8) | 13 (65.0) | 1 | 0.7409 | ns | ns |
| 2DL2-029.1 | 10 (55.6) | 10 (58.8) | 19 (95.0) | 1 | 0.7409 | ns | ns |
| 2DL3-003.1 | 15 (83.3) | 14 (82.4) | 0 (0) | 1 | 0.3282 | ns | ns |
| 2DL3-027.1 | 0 (0) | 3 (17.6) | 20 (100) | 0.1039 | 1 | ns | ns |
| 2DL4-004.1 | 18 (100) | 17 (100) | 11 (55.0) | 1 | 1 | ns | ns |
| 2DL5-005.1 | 12 (66.7) | 11 (64.7) | 8 (40.0) | 1 | 0.5216 | ns | ns |
| 2DL5-006.1 | 8 (44.4) | 9 (52.9) | 7 (35.0) | 0.7395 | 1 | ns | ns |
| 2DL5-033.1 | 9 (50.0) | 8 (47.1) | 0 (0) | 1 | 0.5118 | ns | ns |
| 2DL5-023.2 | 0 (0) | 0 (0) | 2 (10.0) | 1 | 1 | ns | ns |
| 2DL5-024.1 | 0 (0) | 0 (0) | 20 (100) | 1 | 0.4879 | ns | ns |
| 2DP1-025.1 | 16 (88.9) | 16 (94.1) | 7 (35.0) | 1 | 0.2176 | ns | ns |
| 2DS1-008.1 | 12 (66.7) | 10 (58.8) | 11 (55.5) | 0.7332 | 0.1031 | ns | ns |
| 2DS2-009.2 | 10 (55.6) | 10 (58.8) | 8 (40.0) | 1 | 1 | ns | ns |
| 2DS3-010.2 | 5 (27.8) | 4 (23.5) | 18 (90.0) | 1 | 0.5064 | ns | ns |
| 2DS4-012.1 | 14 (77.8) | 14 (82.4) | 10 (50.0) | 1 | 0.3945 | ns | ns |
| 2DS4-026.1 | 6 (33.3) | 8 (47.1) | 8 (40.0) | 0.4998 | 0.3423 | ns | ns |
| 2DS5-013.1 | 9 (50.0) | 8 (47.1) | 20 (100) | 1 | 0.7446 | ns | ns |
| 3DL1-014.1 | 18 (100) | 15 (88.2) | 7 (35.0) | 0.2286 | 1 | ns | ns |
| 3DL1-020.1 | 6 (33.3) | 4 (23.5) | 10 (50.0) | 0.7112 | 1 | ns | ns |
| 3DL1-032.1 | 11 (61.1) | 7 (41.2) | 20 (100) | 0.3175 | 0.5318 | ns | ns |
| 3DL1-031.1 | 18 (100) | 15 (88.2) | 20 (100) | 0.2286 | 1 | ns | ns |
| 3DL2-015.1 | 18 (100) | 17 (100) | 20 (100) | 1 | 1 | ns | ns |
| 3DL3-016.1 | 18 (100) | 17 (100) | 20 (100) | 1 | 1 | ns | ns |
| 3DP1-017.1 | 16 (88.9) | 16 (94.1) | 5 (25.0) | 1 | 0.2176 | ns | ns |
| 3DP1-018.1 | 6 (33.3) | 8 (47.1) | 8 (40.0) | 0.4998 | 0.7240 | ns | ns |
| 3DS1-019.1 | 10 (55.6) | 9 (52.9) |  | 1 | 0.5160 | ns | ns |

n: Number of patients.

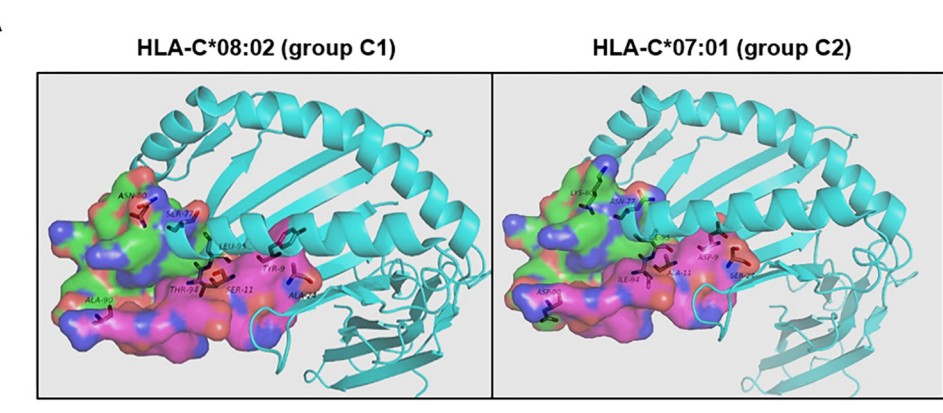

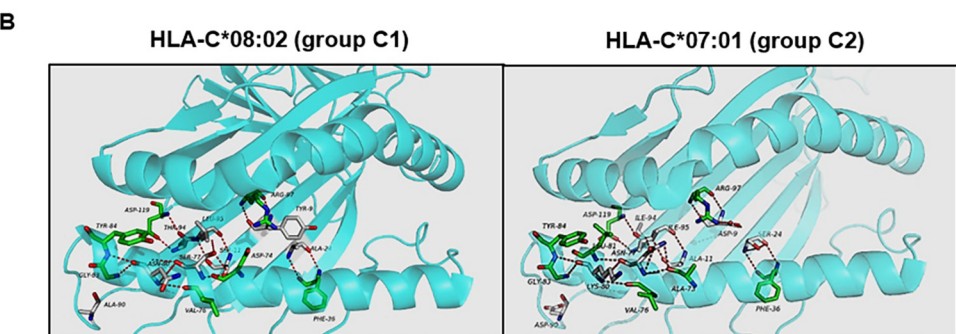

**Fig 1. Comparative structure of peptide-binding pockets of HLA-C1 and -C2 alleles.** (A) 3D modelling of HLA-C\*07:02 (C1) as a template to perform mutagenesis in silico in positions 9, 11, 24, 77, 80, 90, 94, and 95 (grey residues) of the binding pocket, according to the amino acids presented by HLA-C\*08:02 allele (group C1) (Tyr9, Ser11, Ala24, Ser77, Asn80, Ala90, Thr94, and Leu95) (left image) or by HLA-C\*07:01 allele (group C2) (Asp9, Ala11, Ser24, Asn77, Lys80, Asp90, Ile94, and Ile95) (right image). (B) Hydrogen-type bonds between amino acids are represented with red lines and both residue-residue and residue-antigenic amino acids interactions in the pocket are represented in green.

individuals from our cohort, we did not find statistical significance in the comparison between groups (S2 Fig).

In order to determine whether the amino acid exchanges at positions 9, 11, 24, 77, 80, 90, 94, and 95, which are characteristic of groups C1 and C2, may differentially influence the binding-pocket structure in HLA-C alleles, we used the 3D structure of HLA-C\*07:02 [53] as a template to perform mutagenesis in silico in these positions (marked as grey residues in Fig 1A and 1B). These positions were changed to Tyr$^9$, Ser$^{11}$, Ala$^{24}$, Ser$^{77}$, Asn$^{80}$, Ala$^{90}$, Thr$^{94}$, and Leu$^{95}$, as are in HLA-C\*08:02 allele, which belongs to group C1 and its frequency was significantly higher in individuals with mild COVID-19 (Fig 1A, left image), as well as to Asp$^9$, Ala$^{11}$, Ser$^{24}$, Asn$^{77}$, Lys$^{80}$, Asp$^{90}$, Ile$^{94}$, and Ile$^{95}$, as are in HLA-C\*07:01 allele, which belongs to group C2 and its frequency was significantly higher in individuals with severe COVID-19 (Fig 1A, right graph). We observed that these amino acid exchanges induced a conformational modification in the binding pocket structure due to changes in the number and/or position of hydrogen-type bonds (red lines), which could affect both residue-residue and residue-antigenic amino acids interactions in the pocket (green residues) (Fig 1B), as well as the molecule stability.

The combination between HLA-C alleles and their specific KIR alleles was also analyzed in our cohort and no statistically significant differences were found in the comparison between groups (S3 Fig).

## SARS-CoV-2 epitope analysis

Inter-allelic differences associated with COVID-19 severity may reflect differences in the whole number of SARS-CoV-2-epitope repertoires. Therefore, epitopes for each significant HLA class I allele were predicted and further verified if they were associated with experimentally confirmed ligands, as described above (see Methods). Since presentation of HLA class I epitopes is largely determined by protein amount [54], the analysis was focused in the three SARS-CoV-2 proteins more abundant: M, N and S [55]. Predictive values reinforced the data observed in this cohort. The HLA-C*12:03 allele, linked to mild COVID-19 presentation in our study, contained the second highest number of verified epitopes, i.e., 65 (p <0.001, Fisher exact test), among 20 HLA-C alleles (26 ± 19 epitopes, mean ± standard deviation) in abundant SARS-CoV-2 proteins. In contrast, HLA-C*04:01 and -C*07:01, both linked to severe and/or critical COVID-19 presentations in our cohort, ranked 16 and 17th on this scale with only eight of these epitopes each (p = 0.0030). Conflicting results were obtained for HLA-B*14:01, with only one validated abundant epitope, and for HLA-C*08:02, with two epitopes, despite these alleles were also linked to mild symptoms.

## Discussion

Spain was one of the most severely affected European countries during the first pandemic peak of COVID-19 that occurred between March and June 2020 [56]. At this time, the mortality rate during hospitalization was approximately 30%, and even higher after the admission at the ICU, which was approximately 50% [57]. The underlying mechanisms that influenced the poor prognosis of COVID-19 in Spain during this first pandemic wave are still not fully understood. It is already known that some host factors are essential to conditionate the development of a severe or critical presentation of the disease, such as male gender, to be over 65 years of age, and to have previous comorbidities [58]. However, other host factors could also be responsible for the development of an exacerbated inflammatory response [59] that was ineffectual to clear the infection due to an impaired cytotoxic response [8].

The HLA complex has an essential role in the development of an effective immune response against pathogens, thereby influencing the clinical outcomes of infectious diseases, such as the ones caused by human coronaviruses [39–41,60]. The high polymorphism presented by HLA molecules may help explain the great variability of immune responses that are developed by the individuals within a community, including the higher susceptibility to progress to a critical form of COVID-19. In fact, the lower incidence of SARS-CoV-2 in Africa has been related to genetic factors such as the occurrence of different HLA alleles in comparison with other regions [61]. Therefore, the variability in HLA may be also considered an essential host factor that impacts the progression of COVID-19.

In this pilot study, we analyzed whether the presence of specific HLA molecules may influence the different outcomes of COVID-19 in a cohort of individuals with Spanish Mediterranean Caucasian ethnicity who developed different forms of the disease at the beginning of the pandemic in 2020. The Spanish population presents distinctive allelic distributions and haplotypic associations that are consistent with multiple migrations of different ethnicities that occurred throughout history in Spain [62] and that show a relative homogenous distribution fitting the HWE model [27]. According to previous reports [63], most individuals of our cohort who were hospitalized with severe and critical COVID-19 were males (63.5%) with more than 65-year-old, whereas most individuals who developed asymptomatic or mild COVID-19 were females (61.5%) with 47.5-year-old on average.

As expected, HLA-B was the locus that showed the highest number of alleles in our cohort [27]. The presence of specific HLA-B alleles has been associated with increased susceptibility

to different viral infections and therefore, it may be considered a potential predictive bio-marker of COVID-19 severity [32,64]. However, although we found HLA-B rare alleles which presence was significantly different between the groups of individuals with mild and severe presentations of COVID-19 (p = 0.0080; pc = 0.0400), this significance was not extended to individuals with critical COVID-19, which reduced its biological importance. On the other hand, HLA-A*02:01 was the most common allele in all groups of our cohort, which was expected for a population of Spanish origin [49,65], but no significant differences were found in the comparisons of HLA-A alleles frequencies between the groups with different presenta-tions of COVID-19. Regarding HLA-C locus, it showed the highest significance in the different frequency between groups, being HLA-C*04:01 and -C*05:01 the most common HLA-C alleles in all groups. The frequency of HLA-C*08:02 was significantly higher in individuals of our cohort with mild COVID-19 (18.8%) in comparison with individuals with critical COVID-19 (1.6%; p = 0.0024; pc = 0.0240). The frequency of this allele HLA-C*08:02 in the Spanish population is 5.3% [27], which meant that it was 3.5-fold more prevalent in the group of individuals with mild COVID-19 of our cohort. Moreover, 43.8% of the individuals with mild COVID-19 presented an association of HLA-C*08:02 or -C*12:03 or -C*16:01 alleles, all of them classified as alleles from group C1, which was 2.4-fold more frequent than in the gen-eral Spanish population. In comparison, individuals with severe and critical COVID-19 showed a reduced frequency of the associations between these alleles, being 8.3% (p = 0.0030; pc = 0.033) and 16.1% (p = 0.0014; pc = 0.0154), respectively. This may indicate a protective effect of alleles from group C1 against the infection by SARS-CoV-2. Moreover, due to the close distance between HLA-B and -C loci, their expression may be considered together [66] and accordingly, B*14:01-C*08:02 haplotype was one of the most frequent extended haplotypes in the group of mild COVID-19, as well as in the general Spanish population [27,67]. In fact, B*14:01-C*08:02 haplotype was 8.6-fold less frequent in individuals with critical COVID-19, in comparison with the group with mild disease, which also may indicate a potential protective role for this extended haplotype. Although the presence of C2 alleles such as HLA-C*06 or -C*07 [29] and, specifically HLA-C*07:01 or -C*06:02, was significantly increased in individu-als with severe COVID-19 (35.3%) in comparison with participants with mild COVID-19 (6.3%; p = 0.0012; pc = 0.0132), this difference was not significant in the comparison between individuals with critical and mild COVID-19.

The protection exerted by specific HLA alleles may be related to a more efficient antigenic presentation to T lymphocytes, but also to an effective NK cell activation. The activation of NK cells is modulated by a subtle balance between stimulatory and inhibitory signals that are received by several receptors expressed on the cell surface, including KIRs [68]. Consequently, the interaction between HLA and KIR molecules may influence NK cell activation and, conse-quently, the clearance of viral infections [69]. In this regard, previous reports predict that some C2 alleles such as HLA-C*06:02 and–C*07:01 would bind weakly to SARS-CoV-2 antigens, allowing a poorer disease progression [70], most likely due to a deficient antigen presentation and NK cell activation. These results are in accordance with our prediction that C2 alleles such as HLA-C*07:01 and -C*04:01 would be able to present a reduced number of epitopes from the most abundant SARS-CoV-2 proteins M, N, and S. However, these models need to be con-firmed with functional analyses due to they may not always provide confirmatory results. This is the case of C1 alleles such as HLA-C*08 and-C*12, which have also been related to a better cytotoxic response against SARS-CoV-2 [70] and, accordingly, the presence of HLA-C*08:02 and -C*12:03 was significantly higher in individuals with mild COVID-19 from our cohort. Our predictive SARS-CoV-2 epitope analyses confirmed that HLA-C*12:03 allele was able to present a high number of verified epitopes, whereas it could not be confirmed for

HLA-C*08:02, at least for the epitopes from proteins M, N, and S. Therefore, we cannot rule out that HLA-C*08:02 may present efficiently epitopes from other less abundant proteins.

The HLA-C molecules may participate in the activation or inhibition of NK cells by acting as ligands for KIRs [25]. To our knowledge, the contribution of the association between specific HLA-C alleles and KIRs to the clinical presentations of COVID-19 has not been determined yet. Peptides presented by HLA-C alleles from group C1 that shows Asn80 in the heavy chain (HLA-C$^{Asn80}$) would modulate the activation of NK cells through the interaction with inhibitory KIR2DL2 and KIR2DL3, but also with the activating KIR2DS2 [71]. Due to C1 alleles HLA-C*08:02, -C*12:03, and -C*16:01 were significantly more frequent in the group with mild COVID-19 and they have been associated to a protective effect during COVID-19, this may suggest that NK cells from these patients could be more efficient to clear the infection by SARS-CoV-2. In fact, a higher presence of C1 alleles has also been described in patients able to clear spontaneously the infection with hepatitis C virus [72]. On the contrary, the presence of C2 alleles (HLA-C$^{Lys80}$) such as HLA-C*07:01 and -C*06:02, which are ligands for the inhibitory KIR2DL1 and activating KIR2DS1 [30], would theoretically lead to lower NK inhibition and higher activation would be more beneficial for the clearance of viral infections [73]. Therefore, we hypothesized that KIR2DL1/HLA-C group C2 interactions would confer strong inhibitory responses [74,75], at least in those individuals with critical COVID-19 from our cohort that were homozygous for HLA-C group C2 alleles and of whom we previously described that they show an impaired cytotoxic response against SARS-CoV-2 mostly based on high levels of NK, NKT, and CD8+ T cells that display immune exhaustion markers and poor cytotoxic functionality [8].

We also determined that changes of amino acid residues at essential positions in the binding pocket of HLA-C alleles of group C1 (HLA-C*08:02, which was significantly more expressed in individuals with mild COVID-19) and C2 (HLA-C*07:01) modified the groove structure, changing the positions of hydrogen-type bonds and amino acids interactions within residues between both molecules. All these changes would undoubtedly affect the molecule stability, thereby influencing the quality of the immune response induced by these molecules. Moreover, it has been described that alleles of group C1 such as HLA-C*08 display greater stability than alleles of group C2 such as HLA-C*07 [76], most likely due to the reduced ability of HLA-C*07 to present a large range of peptides, which is essential to stabilize the trimeric complexes of HLA-C/β2M/peptide on the cell surface [77]. In fact, HLA-C*07 is considered a low expressed variant with low β2M/peptide binding stability [76] and individuals with this HLA-C allotype usually present cells that express in the surface a large pool of HLA-C free heavy chains, unable to present peptides to the T cells [78]. On the contrary, individuals with more stable HLA-C allotypes such as HLA-C*08 present cells that express on the surface a greater quantity of HLA-C/β2M/peptide heterotrimers able to induce an efficient antiviral response. Therefore, the presence of more stable HLA-C alleles of group C1 in individuals with mild COVID-19 could contribute to their more efficient immune response, which was characterized by a higher cytotoxic activity against SARS-CoV-2 infected cells [8].

In conclusion, the great variability in the clinical presentations of COVID-19 appears to be consequence of several host factors, such as age, gender, and the presence of comorbidities. However, other factors such as an impaired ability to present viral peptides to the immune cells due to unstable peptide-HLA molecule conformations may also be considered, mostly because an impaired cellular immune response has also been related with a poor prognosis during COVID-19. In this observational pilot study, we determined that a small cohort of Spanish Mediterranean Caucasian individuals with a mild or asymptomatic presentations of COVID-19 expressed HLA alleles with potentially protective effect such as highly stable HLA-C alleles of group C1 such as HLA-C*08:02, -C*12:03, and -C*16:01. On the contrary,

47% of individuals with critical COVID-19 were homozygous for HLA-C2 alleles, which are more unstable and could be associated to a poorer disease prognosis due to their lower efficiency to present SARS-CoV-2 peptides to CD8+ T cells and/or activate NK cells.

One main limitation of our study was the relatively small size of the cohort. Despite of this, we observed statistical significance in the comparisons between groups about the frequency of the associations between HLA-C alleles of group C1. In addition, the size of our cohort is similar to other studies previously published [44,45,79,80] that also validate a correlation between HLA-C and an increased COVID-19 severity in the Spanish population [44,45]. Therefore, these data may help us to get further understanding of the underlying mechanisms of COVID-19 pathogenesis, although they must be taken with caution until they are confirmed in a larger cohort. Importantly, to our knowledge, this is the first study that describes a relationship between the presence of specific HLA-C alleles and the different presentations and prognosis of COVID-19. Further analyses will be necessary to determine the actual role of HLA-C locus as a predictive biomarker for COVID-19 progression.

## Supporting information

**S1 Fig. Distribution of Bw4 and Bw6 genotypes in our cohort of individuals with COVID-19.** HLA-B alleles are divided into HLA-Bw6 (Ser77 and Asn80) and -Bw4 (characterized by at least seven different patterns of amino acid changes at positions 77 and 80–83), based on their serological reaction. Statistical significance was calculated using Fisher's exact test. Significance levels were corrected by Bonferroni correction.
(TIF)

**S2 Fig. Analysis of the frequency of HLA-C1 and -C2 alleles in the cohort.** Distribution of alleles from HLA-C1 (HLA-C Asn80) and -C2 (HLA-C Lys80) groups in the individuals with mild, severe and critical COVID-19. The numbers above the bars indicate the percentage of HLA-C alleles' frequency for each group of individuals. The statistical significance of the comparisons between the allele frequencies of groups of individuals with severe and critical COVID-19 and the group of individuals with mild COVID-19 was calculated using Fisher's exact test 2x2. Significance levels were corrected by Bonferroni correction.
(TIF)

**S3 Fig. Interactions between HLA-CAsn80 and HLA-CLys80 and specific KIR alleles.** Interactions between HLA-C alleles from group C1 (HLA-CAsn80) and KIR2DL2, KIR2DL3 and KIR2DS2 (A), or between HLA-C alleles from group C2 (HLA-CLys80) and KIR2DL1 and KIR2DS1 (B) were evaluated in individuals with different presentations of COVID-19. Statistical significance was calculated using Fisher's exact test. Significance levels were corrected by Bonferroni correction.
(TIF)

**S1 Table. Demographic and clinical characteristics of homebound patients with mild COVID-19 who were recruited for this study at the Primary Healthcare Center Laín Entralgo (Alcorcón, Madrid, Spain).**
(DOCX)

**S2 Table. Demographic and clinical characteristics of hospitalized patients with severe and critical COVID-19 who were recruited for this study at the Hospital Universitario Ramon y Cajal, Hospital Universitario Puerta de Hierro and Hospital de El Escorial (Madrid, Spain).**
(DOCX)

## Acknowledgments

We greatly appreciate all the individuals who participate in this study. We thank Daniel Fuertes (School of Telecommunications Engineering, Universidad Politécnica, Madrid, Spain) and Dr Amanda Fernández-Rodriguez (National Center of Microbiology, Instituto de Salud Carlos III, Madrid, Spain) for the revision of all statistical analyses performed in this study. We also thank the excellent secretarial assistance of Ms. Olga Palao. The full membership list of the Contributing members of the Multidisciplinary Group of Study of COVID-19 (MGS-COVID) is as follows:

[1]Centro de Salud Doctor Pedro Laín Entralgo, Alcorcón, Spain

[2]Hematology Service, Hospital Universitario Ramón y Cajal, Madrid, Spain

[3]Infectious Diseases Service, Hospital Universitario Ramón y Cajal, Madrid, Spain

[4]Neumology Service, Hospital Universitario Puerta de Hierro, Majadahonda, Spain

[5]Intensive Medicine Service, Hospital Universitario Ramón y Cajal, Madrid, Spain

#Contributing members of the Multidisciplinary Group of Study of COVID-19 (MGS-COVID) (in alphabetical order):

Esther Alonso Herrador[1], Pablo Amich Alemany[1], Victoria Bosch Martos[1], Sandra Chamorro[2], Belén Comeche[3], Lorena Cordova Castaño[1], Susana Domínguez-Mateos[1], Aurora Expósito Mora[1], Valle Falcones[4], María Mercedes Gea Martinez[1], Alberto Gomez Bonilla[1], María Victoria Leon Gomez[1], Gema Lora Rey[1], Alejandro Luna de Abia[2], Patricia Mínguez[4], Maria Luisa Muñoz Balsa[1], Javier Pérez Gonzalez[1], Sandra Pérez-Santos[1], Jose Sanchez Hernández[1], Cruz Soriano[5], Andrea Vinssac Rayado[1].

## Author Contributions

**Conceptualization:** María Rosa López-Huertas, Mayte Coiras.

**Formal analysis:** Lorena Vigón, Miguel Galán, Jorge Martínez-Laso.

**Funding acquisition:** Mayte Coiras.

**Investigation:** Lorena Vigón, Elena Mateos.

**Methodology:** Lorena Vigón, Miguel Galán, Montserrat Torres, Antonio J. Martín-Galiano, Sara Rodríguez-Mora, Elena Mateos, Magdalena Corona, Rosa Malo, Cristina Navarro, María Aránzazu Murciano-Antón, Valentín García-Gutiérrez, Vicente Planelles, Jorge Martínez-Laso, María Rosa López-Huertas.

**Software:** Antonio J. Martín-Galiano, Jorge Martínez-Laso.

**Writing – review & editing:** Mayte Coiras.

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
