## [Decision Letter · Decision Letter 0]

16 May 2022

PONE-D-22-11703Association between HLA-C alleles and COVID-19 severity in a pilot study with a Spanish Mediterranean Caucasian cohortPLOS ONE

Dear Dr. Vigon,

Thank you for submitting your manuscript to PLOS ONE. After careful consideration, we feel that it has merit but does not fully meet PLOS ONE’s publication criteria as it currently stands. Therefore, we invite you to submit a revised version of the manuscript that addresses the points raised during the review process.

    As you may note reviewr one had some serious concerns and along with reviewer 2 has made a major issue with regards to the small sample size and the requirment for using appropriate statistics for the analysis of data. Kindly, please pay attention to each of the issues raised by reviewer one and the issue raised by reviewer two.

We look forward to receiving your revised manuscript.

Kind regards,

Aftab A. Ansari, PhD

Academic Editor

PLOS ONE

Journal Requirements:

2. One of the noted authors is a group or consortium "Multidisciplinary Group of Study of COVID-19 (MGS-COVID)". In addition to naming the author group, please list the individual authors and affiliations within this group in the acknowledgments section of your manuscript. Please also indicate clearly a lead author for this group along with a contact email address.

5. Please upload a copy of Supporting Information Tables 1 and 2 which you refer to in your text on page 13.

Reviewers' comments:

Reviewer's Responses to Questions

**Comments to the Author**

1. Is the manuscript technically sound, and do the data support the conclusions?

Reviewer #1: No

Reviewer #2: Partly

2. Has the statistical analysis been performed appropriately and rigorously? 

Reviewer #1: No

Reviewer #2: No

3. Have the authors made all data underlying the findings in their manuscript fully available?

Reviewer #1: Yes

Reviewer #2: Yes

4. Is the manuscript presented in an intelligible fashion and written in standard English?

Reviewer #1: No

Reviewer #2: Yes

5. Review Comments to the Author

Reviewer #1: Reviewer comments

The authors conducted HLA and KIR association studies in 72 individuals with Spanish Mediterranean Caucasian ethnicity

1. There are a couple of previously published articles focusing on the same population group, please kindly describe these studies results in comparison to the authors results in both the introduction and the discussion part of the paper.

2. Please describe the clinical protocols that the authors used for the defining of mild, severe and critical COVID-19 patients, did the authors follow the WHO guideline for example SpO2 level, etcs?

3. Please perform a multiple correction adjustment for all the p-values in the HLA association analysis, for example the HLA-B*14:01 would not be significant after multiple correction; P=0.023*12, thus there are no HLA alleles significantly associate with the severity of COVID-19 and the title and the writings of the paper should reflect as such.

4. The obvious disadvantage of this paper are the tiny sample size used in this study, the lack of healthy control, the lack of replication sample set, the lack of appropriate adjustment for multiple testing.

5. The authors performed extra HLA association test by HLA carrier frequencies, but most of the significant alleles are either very rare (example: less than 5), these results should be gathered and put in a combined group such as the rare HLA group.

Minor comments

1. “Most of these alleles concentrate the highest rate of mutations in the exon

that encodes the peptide-binding pocket, likely due to the selection pressure exerted by

infectious diseases during pandemics. “ the reason why mutations have been reported in the exon are mainly caused by the fact that conventional HLA genotyping method such as PCR-SSOP often only targets exon2.3 of the HLA genes, causing a bias in the reporting of exonic variants in the IMGT/HLA database, in fact, the recent database is now full of intronic/3’UTR/5’UTR variants.

Reviewer #2: Overall this is a very interesting study.

I would have one major comment, before any interpretation of the data is possible. I could not find the p-values adjusted for multiple comparisons. Please add

6. PLOS authors have the option to publish the peer review history of their article (what does this mean?). If published, this will include your full peer review and any attached files.

Reviewer #1: No

Reviewer #2: No

---

## [Author Response · Author response to Decision Letter 0]

23 Jun 2022

Answers to Reviewer#1:

Q1. The authors conducted HLA and KIR association studies in 72 individuals with Spanish Mediterranean Caucasian ethnicity 1. There are a couple of previously published articles focusing on the same population group, please kindly describe these studies results in comparison to the authors results in both the introduction and the discussion part of the paper.

A1. We thank the Reviewer#1 for this comment. We had already included two references (44 and 45) published by Iturrieta-Zuazo I, et al. 2020 and Lorente L, et al. 2021 that describe the frequency of HLA alleles in the Spanish population. The sentence “Other HLA-C alleles from HLA-C1 group have been described to show a higher degree of recognition of SARS-CoV-2 peptides (Iturrieta-Zuazo I, et al. 2020), and HLA-C*01 has also been correlated with a higher mortality during COVID-19 (Lorente L, et al. 2021)” had been included in line 154, page 8 (line 149, page 7 in the Revised version). We also discussed these references in lines 643-646, and we have added one sentence highlighted in red: “…it is similar to other studies previously published (44, 45, 82, 83) that also validate a correlation between HLA-C and higher COVID-19 severity in the Spanish population (44, 45).”

Q2. Please describe the clinical protocols that the authors used for the defining of mild, severe and critical COVID-19 patients, did the authors follow the WHO guideline for example SpO2 level, etcs?

A2. We agree with Reviewer#1 that this is an essential question. Patients were classified depending on the medical assistance they received and according to the progression of the disease. Mild COVID-19 received primary health assistance from Primary Healthcare Centers, whereas severe and critical COVID-19 were hospitalized. Critical COVID-19 were admitted to the Intensive Care Unit (ICU). This information was included in lines 279-289. This classification was also based on the WHO’s recommendations (World Health Organization. (2021). Living guidance for clinical management of COVID-19 [Ebook]. Retrieved from https://www.who.int/publications/i/item/WHO/2019-nCoV/clinical/2021.2). We also include Tables S1 and S2 to clarify the data.

Q3. Please perform a multiple correction adjustment for all the p-values in the HLA association analysis, for example the HLA-B*14:01 would not be significant after multiple correction; P=0.023*12, thus there are no HLA alleles significantly associate with the severity of COVID-19 and the title and the writings of the paper should reflect as such.

A3. We thank Reviewer#1 for this critical observation. We agree with the necessity of performing this analysis and therefore, P values have been corrected using Bonferroni correction after alleles with an expected value <5 in each group had been grouped as “rare alleles”. For each HLA haplotype the corrected P value (Pc) was calculated by multiplying the P value obtained with the two-tailed Chi-square or Fisher’s exact test by the number of tested allelic combinations. Only Pc values lower than 0.05 were considered to be statistically significant. Significant differences regarding HLA-C*08:02 are maintained between Mild and Critical COVID-19 (pc=0.0240) (Table 2). Similarly, HLA-C*08:02 or HLA-C*12:03 or HLA-C*16:01 are significantly more frequent in Mild group than in Severe (pc=0.033) and Critical (pc=0.0154) groups (Table 3). In addition, HLA-C*06:02 or HLA-C*07:01 are significantly more prevalent in the Severe group regarding Mild COVID-19 (pc=0.0132) (Table 3). After Bonferroni correction, no statistical significance was observed in HLA haplotypes (Table 4) or in allele HLA-B*14:01. We have modified the manuscript accordingly, adding several sentences highlighted in red throughout the text.

Q4. The obvious disadvantage of this paper are the tiny sample size used in this study, the lack of healthy control, the lack of replication sample set, the lack of appropriate adjustment for multiple testing.

A4. We understand the concern raised by Reviewer#1 about this limitation of our study. We previously discuss this issue in lines 645-652. We would like to indicate, first, that we attained statistical significance in some comparison, despite the reduced sample size. And second, that other studies that evaluate the association of HLA with COVID-19 severity is a small sample have been published previously (Iturrieta-Zuazo I, et al. 2020; Lorente L, et al. 2021; Wang W, et al 2020; Abdelhafiz AS, et al.2021). Besides, these cohorts of individuals with COVID-19 pre-vaccination era are very precious as this situation is unlikely to be repeated again, due to more than 95% of the Spanish population and 65% of world population have at least received one dose of the authorized vaccines. Finally, we also indicated in the manuscript that this observational study is consider a pilot study, and the results would need to be confirmed in a larger cohort.

The aim of this study was to find a possible association between HLA alleles and COVID-19 severity. Because of that, a healthy group was not considered, similarly to previous reports with a similar objective (Iturrieta-Zuazo I, et al. 2020). Moreover, the frequency of the different HLA alleles in the Spanish population have been mentioned in the introduction (Montero-Martín G, et al.2019) (Lines 123-124) and discussed (Montero-Martín G, et al.2019; Arnaiz-Villena A, et al. 1997. Martinez-Laso J, et al. 1995) lines 521-524; 529-530; 538-539; 546-552; 553-556). 

We also agree with Reviewer#1 that the lack of multiple adjustment was necessary to obtain significant results. According to Reviewer#1 suggestion, we applied Bonferroni correction to adjust p values, as explained above.

Q5. The authors performed extra HLA association test by HLA carrier frequencies, but most of the significant alleles are either very rare (example: less than 5), these results should be gathered and put in a combined group such as the rare HLA group.

A5. We agree with Reviewer#1 that this suggestion is also necessary to clarify data expressed in Table 2 and also contributes to get a better understanding of the results. Consequently, those alleles with an expected value <5 in each group have been grouped as “rare alleles” for every HLA class I and class II genes tested. Next, Bonferroni correction was performed, considering this new group of “rare alleles” as one determination.

Minor comments

Q1. “Most of these alleles concentrate the highest rate of mutations in the exon that encodes the peptide-binding pocket, likely due to the selection pressure exerted by infectious diseases during pandemics“. The reason why mutations have been reported in the exon are mainly caused by the fact that conventional HLA genotyping method such as PCR-SSOP often only targets exon2.3 of the HLA genes, causing a bias in the reporting of exonic variants in the IMGT/HLA database, in fact, the recent database is now full of intronic/3’UTR/5’UTR variants.

A1. We thank Reviewer#1 for this suggestion. In order to avoid misunderstandings, we believe that is better to delete this sentence.

 

Answers to Reviewer#2:

Q1. I would have one major comment, before any interpretation of the data is possible. I could not find the p-values adjusted for multiple comparisons. Please add.

A1. We thank Reviewer#1 for this suggestion. We agree with the necessity of performing this analysis and therefore, P values have been corrected using Bonferroni correction after alleles with an expected value <5 in each group had been grouped as “rare alleles”. For each HLA haplotype the corrected P value (Pc) was calculated by multiplying the P value obtained with the two-tailed Chi-square or Fisher´s exact test by the number of tested allelic combinations. Only Pc values lower than 0.05 were considered to be statistically significant. Significant differences regarding HLA-C*08:02 are maintained between Mild and Critical COVID-19 (pc=0.0240) (Table 2). Similarly, HLA-C*08:02 or HLA-C*12:03 or HLA-C*16:01 are significantly more frequent in Mild group than in Severe (pc=0.033) and Critical (pc=0.0154) groups (Table 3). In addition, HLA-C*06:02 or HLA-C*07:01 are significantly more prevalent in the Severe group regarding Mild COVID-19 (pc=0.0132) (Table 3). After Bonferroni correction, no statistical significance was observed in HLA haplotypes (Table 4) or in allele HLA-B*14:01. We have modified the manuscript accordingly, adding several sentences highlighted in red throughout the text.

---

## [Editor Report · Decision Letter 1]

28 Jul 2022

Association between HLA-C alleles and COVID-19 severity in a pilot study with a Spanish Mediterranean Caucasian cohort

PONE-D-22-11703R1

Dear Dr. Coiras,

We’re pleased to inform you that your manuscript has been judged scientifically suitable for publication and will be formally accepted for publication once it meets all outstanding technical requirements.

Kind regards,

Gualtiero I. Colombo, M.D., Ph.D.

Academic Editor

PLOS ONE
---

## [Editor Report · Acceptance letter]

3 Aug 2022

PONE-D-22-11703R1 

Association between HLA-C alleles and COVID-19 severity in a pilot study with a Spanish Mediterranean Caucasian cohort 

Dear Dr. Coiras:

I'm pleased to inform you that your manuscript has been deemed suitable for publication in PLOS ONE. Congratulations! Your manuscript is now with our production department. 

Kind regards, 

on behalf of

Dr. Gualtiero I. Colombo 

Academic Editor

PLOS ONE